# Learning Deep O($n$)-Equivariant Hyperspheres

## Abstract

This paper presents an approach to learning (deep) $n$D features equivariant under orthogonal transformations, utilizing hyperspheres and regular $n$-simplexes. Our main contributions are theoretical and tackle major challenges in geometric deep learning such as equivariance and invariance under geometric transformations. Namely, we enrich the recently developed theory of steerable 3D spherical neurons— SO(3)-equivariant filter banks based on neurons with spherical decision surfaces— by extending said neurons to $n$D, which we call deep equivariant hyperspheres, and enabling their multi-layer construction. Using synthetic and real-world data in $n$D, we experimentally verify our theoretical contributions and find that our approach is superior to the competing methods for benchmark datasets in all but one case, additionally demonstrating a better speed/performance trade-off in all but one other case.

## 1 Introduction

*Spheres*[1] serve as a foundational concept in Euclidean space while simultaneously embodying the essence of non-Euclidean geometry through their intrinsic curvature and non-linear nature. This motivated their usage as decision surfaces encompassed by spherical neurons (Perwass et al., 2003; Melnyk et al., 2021).

Felix Klein's Erlangen program of 1872 (Hilbert & Cohn-Vossen, 1952) introduced a methodology to unify non-Euclidean geometries, emphasizing the importance of studying geometries through their invariance properties under transformation groups. Similarly, geometric deep learning (GDL) as introduced by Bronstein et al. (2017; 2021) constitutes a unifying framework for various neural architectures. This framework is built from the first principles of geometry—symmetry and scale separation—and enables tractable learning in high dimensions.

Symmetries play a vital role in preserving structural information of geometric data and allow models to adjust to different geometric transformations. This flexibility ensures that models remain robust and accurate, even when the input data undergo various changes. In this context, spheres exhibit a maximal set of symmetries compared to other geometric entities in Euclidean space. The orthogonal group O($n$) fully encapsulates the symmetry structure of an $n$D sphere, including both rotational and reflection symmetries. Integrating these symmetries into a model as an inductive bias is often a crucial requirement for problems in natural sciences and the respective applications, e.g., molecular analysis, protein design and assessment, or catalyst design (Rupp et al., 2012; Ramakrishnan et al., 2014; Townshend et al., 2021; Jing et al., 2021; Lan et al., 2022).

In this paper, we consider data that live in Euclidean space (such as point clouds) and undergo rotations and reflections, i.e., transformations of the O($n$)-group. Enriching the theory of steerable 3D spherical neurons (Melnyk et al., 2022a), we present a method for learning O($n$)-equivariant deep features using regular $n$-simplexes[2] and $n$D spheres, which we call `Deep Equivariant Hyperspheres` (see Figure 1). The name also captures the fact that the vertices of a regular $n$-simplex lie on an $n$D sphere, and that our main result enables combining them in multiple layers, thereby enabling *deep* propagation via them.

Our main contributions are summarized as follows:

---

[1] By *sphere*, we generally refer to an $n$D sphere or a hypersphere; e.g., a circle is thus a 2D sphere.

[2] We use the fact that a regular $n$-simplex contains $n + 1$ equidistant vertices in $n$D.

Figure 1: The central components of `Deep Equivariant Hyperspheres` (best viewed in color): regular $n$-simplexes with the $n$D spherical decision surfaces located at their vertices and the simplex change-of-basis matrices $\mathbf{M}_n$ (displayed for the cases $n = 2$ and $n = 3$).

- We propose a method for learning O($n$)-equivariant deep features, called `Deep Equivariant Hyperspheres`, readily generalizing to any dimension.
- We define and analyze generalized concepts for a network composed of the proposed neurons, such as equivariant bias, non-linearity, and multilayer configuration.
- We provide experiments using both synthetic and real-world data in $n$D and demonstrate the soundness and effectiveness of the developed theoretical framework, achieving state-of-the-art performance in all but one case (CGENN on the O(5) regressions task), and demonstrating a better speed/performance trade-off than the competing methods in all but one other case (VN on the O(3) classification task).

## 2    RELATED WORK

Even though the concept of spheres is also an essential part of spherical convolutional neural networks (CNNs) and CNNs designed to operate on 360 imagery (Coors et al., 2018; Su & Grauman, 2017; Esteves et al., 2018; Cohen et al., 2018; Perraudin et al., 2019), our method does not map input data *on* a sphere, $\mathcal{S}^2$, nor does it perform convolution on a sphere. Instead, it embeds input in a higher-dimensional Euclidean space by means of a quadratic function. Namely, our work extrapolates the ideas from prior work by Perwass et al. (2003); Melnyk et al. (2021), in which spherical decision surfaces and their symmetries have been utilized for constructing equivariant models for the 3D case (Melnyk et al., 2022a;b). We carefully review these works in Section 3.

Similarly to the approach of Ruhe et al. (2023), our `Deep Equivariant Hyperspheres` directly operate on the basis of the input points, not requiring constructing an alternative one, such as a steerable spherical harmonics basis, which is a key limitation of many related methods (Anderson et al., 2019; Thomas et al., 2018; Fuchs et al., 2020). Our method also generalizes to the orthogonal group of any dimensionality.

Another type of method is such as by Finzi et al. (2021), a representation method building equivariant feature maps by computing an integral over the respective group (which is intractable for continuous Lie groups and hence, requires coarse approximation). Another category includes methods operating on scalars and vectors: they update the vector information by learning the parameters conditioned on scalar information and multiplying the vectors with it (Satorras et al., 2021), or by learning the latent equivariant features (Deng et al., 2021).

## 3    BACKGROUND

In this section, we present a comprehensive background on the theory of spherical neurons and their rotation-equivariant version, as well as on the general geometric concepts used in our work.

### 3.1    SPHERICAL NEURONS VIA NON-LINEAR EMBEDDING

Spherical neurons (Perwass et al., 2003; Melnyk et al., 2021) are neurons with, as the name suggests, spherical decision surfaces. By virtue of conformal geometric algebra (Li et al., 2001), Perwass et al. (2003) proposed to embed the data vector $\mathbf{x} \in \mathbb{R}^n$ and represent the sphere with center $\mathbf{c} = (c_1, \ldots, c_n) \in \mathbb{R}^n$ and radius $r \in \mathbb{R}$ respectively as

$$\boldsymbol{X} = \left(x_1, \ldots, x_n, -1, -\frac{1}{2}\|\mathbf{x}\|^2\right) \in \mathbb{R}^{n+2} \quad \text{and} \quad \boldsymbol{S} = \left(c_1, \ldots, c_n, \frac{1}{2}(\|\mathbf{c}\|^2 - r^2), 1\right) \in \mathbb{R}^{n+2}, \quad (1)$$

and used their scalar product $\boldsymbol{X}^\top \boldsymbol{S} = -\frac{1}{2}\|\mathbf{x} - \mathbf{c}\|^2 + \frac{1}{2}r^2$ as a classifier, i.e., the spherical neuron:

$$f_S(\boldsymbol{X}; \boldsymbol{S}) = \boldsymbol{X}^\top \boldsymbol{S}, \tag{2}$$

with learnable parameters $\boldsymbol{S} \in \mathbb{R}^{n+2}$.

The sign of this scalar product depends on the position of the point $\mathbf{x}$ relative to the sphere $(\mathbf{c}, r)$: inside the sphere if positive, outside of the sphere if negative, and on the sphere if zero (Perwass et al., 2003). Geometrically, the activation of the spherical neuron equation 2 determines the cathetus length of the right triangle formed by $\mathbf{x}$, $\mathbf{c}$, and the corresponding point on the sphere (see Figure 2 in Melnyk et al. (2021)).

We note that with respect to the data vector $\mathbf{x} \in \mathbb{R}^n$, a spherical neuron represents a non-linear function $f_S(\,\cdot\,; \boldsymbol{S}) : \mathbb{R}^{n+2} \to \mathbb{R}$, due to the inherent non-linearity of the embedding (equation 1), and therefore, does not necessarily require an activation function, as observed by Melnyk et al. (2021). The components of $\boldsymbol{S}$ in equation 1 can be treated as *independent* learnable parameters. In this case, a spherical neuron learns a *non-normalized* sphere of the form $\widetilde{\boldsymbol{S}} = (s_1, \dots, s_{n+2}) \in \mathbb{R}^{n+2}$, which represents the same decision surface as its normalized counterpart defined in equation 1, thanks to the homogeneity of the embedding (Perwass et al., 2003; Li et al., 2001).

## 3.2 Equi- and invariance under orthogonal transformations

The elements of the orthogonal group $\mathrm{O}(n)$ can be represented as $n \times n$ matrices $\boldsymbol{R}$ with the properties $\boldsymbol{R}^\top \boldsymbol{R} = \boldsymbol{R}\boldsymbol{R}^\top = \mathbf{I}_n$, where $\mathbf{I}_n$ is the identity matrix, and $\det \boldsymbol{R} = \pm 1$, geometrically characterizing $n$D rotations and reflections. The special orthogonal group $\mathrm{SO}(n)$ is a subgroup of $\mathrm{O}(n)$ and includes only orthogonal matrices with the positive determinant, representing rotations.

We say that a function $f : \mathcal{X} \to \mathcal{Y}$ is $\mathrm{O}(n)$-equivariant if for every $\boldsymbol{R} \in \mathrm{O}(n)$ there exists the transformation representation, $\rho(\boldsymbol{R})$, in the function output space, $\mathcal{Y}$, such that

$$\rho(\boldsymbol{R})\, f(\mathbf{x}) = f(\boldsymbol{R}\mathbf{x}) \quad \text{for all } \boldsymbol{R} \in \mathrm{O}(n),\ \mathbf{x} \in \mathcal{X}. \tag{3}$$

We call a function $f : \mathcal{X} \to \mathcal{Y}$ $\mathrm{O}(n)$-invariant if for every $\boldsymbol{R} \in \mathrm{O}(n)$, $\rho(\boldsymbol{R}) = \mathbf{I}_n$. That is, if

$$f(\mathbf{x}) = f(\boldsymbol{R}\mathbf{x}) \quad \text{for all } \boldsymbol{R} \in \mathrm{O}(n),\ \mathbf{x} \in \mathcal{X}. \tag{4}$$

Following the prior work convention (Melnyk et al., 2022a;b) hereinafter, we write $\boldsymbol{R}$ to denote the same $n$D rotation/reflection as an $n \times n$ matrix in the Euclidean space $\mathbb{R}^n$, as an $(n+1) \times (n+1)$ matrix in the projective (homogeneous) space $P(\mathbb{R}^n) \subset \mathbb{R}^{n+1}$, and as an $(n+2) \times (n+2)$ matrix in $\mathbb{R}^{n+2}$. For the latter two cases, we achieve this by appending ones to the diagonal of the original $n \times n$ matrix without changing the transformation itself (Melnyk et al., 2021).

## 3.3 Steerable 3D spherical neurons and TetraSphere

Considering the 3D case, Melnyk et al. (2022a) showed that a spherical neuron (Perwass et al., 2003; Melnyk et al., 2021) can be *steered*. In this context, *steerability* is defined as the ability of a function to be written as a linear combination of the rotated versions of itself, called *basis functions* (Freeman et al., 1991; Knutsson et al., 1992). For details, see Section A in the Appendix.

According to Melnyk et al. (2022a), a 3D steerable filter consisting of spherical neurons needs to comprise a *minimum* of four 3D spheres: one learnable spherical decision surface $\boldsymbol{S} \in \mathbb{R}^5$ (equation 1) and its three copies *rotated* into the other three vertices of the regular tetrahedron, using one of the results of Freeman et al. (1991) that the basis functions must be distributed in the space uniformly.

To construct such a filter, i.e., a steerable 3D spherical neuron, the main (learned) sphere center $\mathbf{c}_0$ needs to be rotated into $\|\mathbf{c}_0\|\,(1, 1, 1)$ by the corresponding (geodesic) rotation $\boldsymbol{R}_O$. The resulting sphere center is then rotated into the other three vertices of the regular tetrahedron. This is followed by rotating all four spheres back to the original coordinate system. One steerable 3D spherical neuron can thus be defined by means of the $4 \times 5$ matrix $B(\boldsymbol{S})$ containing the four spheres:

$$\mathrm{F}(\boldsymbol{X}; \boldsymbol{S}) = B(\boldsymbol{S})\boldsymbol{X}, \quad B(\boldsymbol{S}) = \left[(\boldsymbol{R}_O^\top \boldsymbol{R}_{T_i} \boldsymbol{R}_O\, \boldsymbol{S})^\top\right]_{i=1\dots4}, \tag{5}$$

where $X \in \mathbb{R}^5$ is the input 3D point embedded using equation 1, $\{R_{T_i}\}_{i=1}^4$ is the $\mathbb{R}^5$ rotation isomorphism corresponding to the rotation from the first vertex, i.e., $(1, 1, 1)$ to the $i$-th vertex of the regular tetrahedron[3].

Melnyk et al. (2022a) showed that steerable 3D spherical neurons are SO(3)-equivariant:

$$V_R \, B(S) \, X = B(S) \, RX, \quad V_R = \mathbf{M}^\top R_O \, R \, R_O^\top \mathbf{M} , \tag{6}$$

where $R$ is a representation of the 3D rotation in $\mathbb{R}^5$, and $V_R \in G < \text{SO}(4)$ is the 3D rotation representation in the filter bank output space, with $\mathbf{M} \in \text{SO}(4)$ being a change-of-basis matrix that holds the homogeneous coordinates of the tetrahedron vertices in its columns as

$$\mathbf{M} = \begin{bmatrix} \mathbf{m}_1 & \mathbf{m}_2 & \mathbf{m}_3 & \mathbf{m}_4 \end{bmatrix} = \frac{1}{2} \begin{bmatrix} 1 & 1 & -1 & -1 \\ 1 & -1 & 1 & -1 \\ 1 & -1 & -1 & 1 \\ 1 & 1 & 1 & 1 \end{bmatrix} . \tag{7}$$

We note that with respect to the input vector $\mathbf{x} \in \mathbb{R}^3$, a steerable 3D spherical neuron represents a non-linear rotational-equvivariant function $\text{F}(\,\cdot\,; S) : \mathbb{R}^5 \to \mathbb{R}^4$ with the learnable parameters $S \in \mathbb{R}^5$.

**TetraSphere**   As the first reported attempt to *learn* steerable 3D spherical neurons in an end-to-end approach, Melnyk et al. (2022b) has presently introduced an approach for O(3)-invariant point cloud classification based on said neurons and the VN-DGCNN architecture (Deng et al., 2021), called TetraSphere.

Given the point cloud input $\mathcal{X} \in \mathbb{R}^{N \times 3}$, the TetraSphere approach suggests to learn 4D features of each point by means of the TetraTransform layer $l_{\text{TT}}(\,\cdot\,; S) : \mathbb{R}^{N \times 3} \to \mathbb{R}^{N \times 4 \times K}$ that consists of $K$ steerable spherical neurons $B(S_k)$ (see equation 5) that are shared among the points. After the application of TetraTransform, pooling over the $K$ dimensions takes place, and the obtained feature map is then propagated through the VN-DGCNN network as-is. However, the work of Melnyk et al. (2022b) does not investigate the question of how to combine the steerable neurons in multiple layers, nor how to process data in dimensions higher than 3.

### 3.4    Regular simplexes

Geometrically, a regular $n$-simplex represents $n + 1$ equidistant points in $n$D (Elte, 2006), lying on an $n$D sphere with unit radius. In the 2D case, the regular simplex is an equilateral triangle; in 3D, a regular tetrahedron, and so on.

Following Cevikalp & Saribas (2023), we compute the Cartesian coordinates of a regular $n$-simplex as $n + 1$ vectors $\mathbf{p}_i \in \mathbb{R}^n$:

$$\mathbf{p}_i = \begin{cases} n^{-1/2} \, \mathbf{1}, & i = 1 \\ \kappa \, \mathbf{1} + \mu \, \mathbf{e}_{i-1}, & 2 \le i \le n + 1 \, , \end{cases} \quad \kappa = -\frac{1 + \sqrt{n+1}}{n^{3/2}} \, , \quad \mu = \sqrt{1 + \frac{1}{n}} \, , \tag{8}$$

where $\mathbf{1} \in \mathbb{R}^n$ is a vector with all elements equal to 1 and $\mathbf{e}_i$ is the natural basis vector with the $i$-th element equal to 1.

For the case $n = 3$, we identify the following connection between equation 7 and equation 8: the columns of $\mathbf{M}$, $\mathbf{m}_i \in \mathbb{R}^4$, are the coordinates of the regular 3-simplex appended with a constant and normalized to unit length; that is, $\mathbf{m}_i = \frac{1}{p} \begin{bmatrix} \mathbf{p}_i \\ 1/\sqrt{3} \end{bmatrix}$ with $p = \left\| \begin{bmatrix} \mathbf{p}_i \\ 1/\sqrt{3} \end{bmatrix} \right\|, 1 \le i \le 4$.

## 4    Deep Equivariant Hyperspheres

In this section, we provide a complete derivation of the proposed O($n$)-equivariant neuron based on a learnable spherical decision surface and multiple transformed copies of it, as well as define and analyze generalized concepts of equivariant bias, non-linearities, and multi-layer setup.

While it is intuitive that in higher dimensions one should use more copies (i.e., vertices) than in the 3D case (Melnyk et al., 2022a), it is uncertain exactly how many are needed. We hypothesize that the vertices should constitute a regular $n$-simplex ($n + 1$ vertices) and rigorously prove it in this section.

---

[3]Therefore, $R_{T_1} = \mathbf{I}_5$, i.e., the original $S$ remains at $\mathbf{c}_0$.

## 4.1 The simplex change of basis

We generalize the change-of-basis matrix (equation 7) to $n$D by introducing $\mathbf{M}_n$, an $(n+1) \times (n+1)$ matrix holding in its columns the coordinates of the regular $n$-simplex appended with a constant and normalized to unit length:

$$\mathbf{M}_n = \big[\mathbf{m}_i\big]_{i=1\ldots n+1}, \quad \mathbf{m}_i = \frac{1}{p}\begin{bmatrix}\mathbf{p}_i \\ n^{-1/2}\end{bmatrix}, \quad p = \left\|\begin{bmatrix}\mathbf{p}_i \\ n^{-1/2}\end{bmatrix}\right\|, \tag{9}$$

where the norms $p$ are constant, since $\|\mathbf{p}_i\| = \|\mathbf{p}_j\|$ for all $i$ and $j$ by definition of a regular simplex.

**Proposition 1.** *Let $\mathbf{M}_n$ be the-change-of-basis matrix defined in equation 9. Then $\mathbf{M}_n$ is an $(n+1)D$ rotation or reflection,* i.e.*, $\mathbf{M}_n \in \mathrm{O}(n+1)$ (see Section B in the Appendix for numeric examples).*

*Proof.* We want to show that $\mathbf{M}_n^\top \mathbf{M}_n = \mathbf{I}_{n+1}$, which will prove that $\mathbf{M}_n$ is orthogonal. The diagonal elements of $\mathbf{M}_n^\top \mathbf{M}_n$ are $\mathbf{m}_i^\top \mathbf{m}_i = \|\mathbf{m}_i\|^2 = 1$ since $\|\mathbf{m}_i\| = 1$. The off-diagonal elements are found as $\mathbf{m}_i^\top \mathbf{m}_j = (\mathbf{p}_i^\top \mathbf{p}_j + n^{-1})/p^2$ for $i \neq j$, where $p$ is defined in equation 9. Note that $\mathbf{p}_i^\top \mathbf{p}_j$ is the same for all $i$ and $j$ with $i \neq j$ since, by definition of a regular simplex, the vertices $\mathbf{p}_i$ are spaced uniformly. Note that $\mathbf{p}_i^\top \mathbf{p}_j = -n^{-1}$ for all $i \neq j$ by definition (equation 8). Hence, the off-diagonal elements of $\mathbf{M}_n^\top \mathbf{M}_n$ are zeros and $\mathbf{M}_n^\top \mathbf{M}_n = \mathbf{I}_{n+1}$. $\square$

Since $\mathbf{M}_n \in \mathrm{O}(n+1)$, the sign of $\det \mathbf{M}_n$ is determined by the number of reflections required to form the transformation. In the case of a regular $n$-simplex, the sign of the determinant depends on the parity of $n$ **and** the configuration of the simplex vertices. In our case, $\mathbf{M}_n$ is a rotation for odd $n$, i.e., $\det \mathbf{M}_n = 1$, and a reflection for even $n$. Consider, for example, the case $n = 3$. The matrix $\mathbf{M}_3$ shown in equation 7 has $\det \mathbf{M}_3 = 1$, thus, is a 4D rotation, as stated in Section 3.3.

**Lemma 2.** *Let $\mathbf{M}_n$ be the change-of-basis matrix defined in equation 9, and $\mathbf{P}_n$ an $n \times (n+1)$ matrix holding the regular $n$-simplex vertices, $\mathbf{p}_i$, in its columns, and $p = \left\|\begin{bmatrix}\mathbf{p}_i \\ n^{-1/2}\end{bmatrix}\right\|$, as defined in equation 9. Then*

$$\mathbf{M}_n \mathbf{P}_n^\top = p\begin{bmatrix}\mathbf{I}_n \\ \mathbf{0}^\top\end{bmatrix}. \tag{10}$$

*Proof.* We begin by elaborating on equation 9:

$$\mathbf{M}_n = \frac{1}{p}\begin{bmatrix}\mathbf{P}_n \\ n^{-1/2}\,\mathbf{1}^\top\end{bmatrix}. \tag{11}$$

We note that the norms of the rows of $\mathbf{P}_n$ are also equal to $p$ since $\mathbf{M}_n \in \mathrm{O}(n+1)$ (as per Proposition 1). Recall that $\mathbf{P}_n$ is centered at the origin, and, therefore, for a constant $a \in \mathbb{R}$ and a vector of ones $\mathbf{1} \in \mathbb{R}^{n+1}$, we obtain $a\,\mathbf{1}^\top \mathbf{P}_n^\top = \mathbf{0}^\top$. Remembering that the product $\mathbf{M}_n \mathbf{P}_n^\top$ is between $\mathbb{R}^{n+1}$ vectors, we plug equation 11 into the LHS of equation 10 and obtain

$$\mathbf{M}_n \mathbf{P}_n^\top = \frac{1}{p}\begin{bmatrix}\mathbf{P}_n \\ n^{-1/2}\,\mathbf{1}^\top\end{bmatrix}\mathbf{P}_n^\top = \frac{p^2}{p}\begin{bmatrix}\mathbf{I}_n \\ \mathbf{0}^\top\end{bmatrix} = p\begin{bmatrix}\mathbf{I}_n \\ \mathbf{0}^\top\end{bmatrix}. \tag{12}$$

$\square$

## 4.2 Equivariant $n$D spheres

In this section, we generalize steerable 3D spherical neurons reviewed in Section 3.3. We denote an equivariant $n$D-sphere neuron (an *equivariant hypersphere*) by means of the $(n+1) \times (n+2)$ matrix $B_n(\boldsymbol{S})$ for the spherical decision surface $\boldsymbol{S} \in \mathbb{R}^{n+2}$ with center $\mathbf{c}_0 \in \mathbb{R}^n$ and an $n$D input $\mathbf{x} \in \mathbb{R}^n$ embedded as $\boldsymbol{X} \in \mathbb{R}^{n+2}$ as

$$\mathbf{F}_n(\boldsymbol{X}; \boldsymbol{S}) = B_n(\boldsymbol{S})\,\boldsymbol{X}, \quad B_n(\boldsymbol{S}) = \Big[(\boldsymbol{R}_O^\top \boldsymbol{R}_{T_i}\,\boldsymbol{R}_O\,\boldsymbol{S})^\top\Big]_{i=1\ldots n+1}, \tag{13}$$

where $\{\boldsymbol{R}_{T_i}\}_{i=1}^{n+1}$ is the $\mathbb{R}^{n+2}$ rotation isomorphism corresponding to the rotation from the first vertex to the $i$-th vertex of the regular $n$-simplex, and $\boldsymbol{R}_O \in \mathrm{SO}(n)$ is the geodesic rotation from the sphere center $\mathbf{c}_0$ to $\|\mathbf{c}_0\|\,\mathbf{p}_1$ (therefore, $\boldsymbol{R}_{T_1} = \mathbf{I}_{n+2}$).

We now need to prove that $\mathbf{F}_n(\,\cdot\,; \boldsymbol{S})$ is $\mathrm{O}(n)$-equivariant.

**Proposition 3.** *Let* $\mathbf{F}_n(\,\cdot\,;\boldsymbol{S}) : \mathbb{R}^{n+2} \to \mathbb{R}^{n+1}$ *be the neuron defined in equation 13 and* $\boldsymbol{R} \in \mathrm{O}(n)$ *be an* $n \times n$ *rotation or reflection matrix. Then the transformation representation in the filter output space* $\mathbb{R}^{n+1}$ *is given by the* $(n+1) \times (n+1)$ *matrix*

$$V_n = \rho\left(\boldsymbol{R}\right) = \mathbf{M}_n^\top \boldsymbol{R}_O \, \boldsymbol{R} \, \boldsymbol{R}_O^\top \mathbf{M}_n \;, \tag{14}$$

*where* $\mathbf{M}_n \in \mathrm{O}(n+1)$ *is the-change-of-basis matrix defined in equation 9 and a 1 is appended to the diagonals of* $\boldsymbol{R}_O$ *and* $\boldsymbol{R}$ *to make them* $(n+1) \times (n+1)$. *Furthermore,* $V_n \in G < \mathrm{O}(n+1)$.

*Proof.* Since $\mathbf{M}_n \in \mathrm{O}(n+1)$, $\boldsymbol{R}_O \in \mathrm{SO}(n)$, and $\boldsymbol{R} \in \mathrm{O}(n)$ are orthogonal matrices, $V_n$ in equation 14 is an orthogonal change-of-basis transformation that represents $\boldsymbol{R} \in \mathrm{O}(n)$ in the basis constructed by $\mathbf{M}_n$ and $\boldsymbol{R}_O$. Note that appending one to the diagonal of $\boldsymbol{R} \in \mathrm{O}(n)$ does not affect the sign of the determinant, which makes $V_n$ a reflection representation if $\det \boldsymbol{R} = -1$, or a rotation representation if $\det \boldsymbol{R} = +1$. Since $\boldsymbol{R} \in \mathrm{O}(n)$ and $\boldsymbol{R}_O \in \mathrm{O}(n)$, not all elements of $\mathrm{O}(n+1)$ can be generated by the operation in equation 14. Thus, we conclude that $V_n$ belongs to a proper subgroup of $\mathrm{O}(n+1)$, i.e., $G < \mathrm{O}(n+1)$. The original transformation is found directly as

$$\boldsymbol{R} = \boldsymbol{R}_O^\top \mathbf{M}_n \, V_n \, \mathbf{M}_n^\top \boldsymbol{R}_O \;, \tag{15}$$

followed by the retrieval of the upper-left $n \times n$ sub-matrix. □

Noteworthy, the basis determined by $\boldsymbol{R}_O \in \mathrm{SO}(n)$, which depends on the center $\mathbf{c}_0$ of the sphere $\boldsymbol{S} \in \mathbb{R}^{n+2}$ (see equation 13), will be different for different $\mathbf{c}_0$. Therefore, the representation $V_n$ will differ as well.

**Theorem 4.** *The neuron* $\mathbf{F}_n(\,\cdot\,;\boldsymbol{S}) : \mathbb{R}^{n+2} \to \mathbb{R}^{n+1}$ *defined in equation 13 is* $\mathrm{O}(n)$*-equivariant.*

*Proof.* To prove the theorem, we need to show that equation 3 holds for $\mathbf{F}_n(\,\cdot\,;\boldsymbol{S})$.

We substitute equation 14 to the LHS and equation 13 to the RHS, and obtain

$$V_n \, B_n(\boldsymbol{S}) \, \boldsymbol{X} = B_n(\boldsymbol{S}) \, \boldsymbol{R} \boldsymbol{X} \;. \tag{16}$$

For the complete proof, please see Section C in the Appendix. □

We note that with respect to the input vector $\mathbf{x} \in \mathbb{R}^n$, the equivariant hypersphere $\mathbf{F}_n(\,\cdot\,;\boldsymbol{S}) : \mathbb{R}^{n+2} \to \mathbb{R}^{n+1}$ represents a non-linear $\mathrm{O}(n)$-equivariant function. It is also worth mentioning that the *sum* of the output $\boldsymbol{Y} = B_n(\boldsymbol{S})\,\boldsymbol{X}$ is an $\mathrm{O}(n)$-invariant scalar, i.e., the DC-component, due to the regular $n$-simplex construction.

This invariant part can be adjusted by adding a scalar *bias* parameter to the output $\boldsymbol{Y}$. The concept of bias is imperative for linear classifiers, but for spherical decision surfaces (Perwass et al., 2003), it is implicitly modeled by the embedding (equation 1). We note, however, that adding a scalar bias parameter, $b \in \mathbb{R}$ to the output of an equivariant hypersphere (equation 13) respects $\mathrm{O}(n)$-equivariance:

**Proposition 5.** *Let* $\boldsymbol{Y} \in \mathbb{R}^{n+1}$ *be the output of the* $\mathrm{O}(n)$*-equivariant hypersphere* $\mathbf{F}_n(\,\cdot\,;\boldsymbol{S}) : \mathbb{R}^{n+2} \to \mathbb{R}^{n+1}$ *(equation 13) given the input* $\boldsymbol{X} \in \mathbb{R}^{n+2}$, *and* $b \in \mathbb{R}$ *be a bias parameter. Then* $\boldsymbol{Y}' = \boldsymbol{Y} + b\,\mathbf{1}$, *where* $\mathbf{1}$ *is the vector of ones in* $\mathbb{R}^{n+1}$, *is also* $\mathrm{O}(n)$*-equivariant.*

*Proof.* We need to show that equation 16 also holds when the bias $b$ is added. First, we use $V_n$—the representation of $\boldsymbol{R} \in \mathrm{O}(n)$ from equation 14—and the fact that $\boldsymbol{R}$ and $\boldsymbol{R}_O$ are both appended 1 to their main diagonal to make them $(n+1) \times (n+1)$. Then $V_n\,\mathbf{1} = \mathbf{M}_n^\top \boldsymbol{R}_O\,\boldsymbol{R}\,\boldsymbol{R}_O^\top \mathbf{M}_n \mathbf{1} = \mathbf{M}_n^\top \boldsymbol{R}_O\,\boldsymbol{R}\,\boldsymbol{R}_O^\top \begin{bmatrix} \mathbf{0} \\ p\sqrt{n} \end{bmatrix} = \mathbf{M}_n^\top \begin{bmatrix} \mathbf{0} \\ p\sqrt{n} \end{bmatrix} = \mathbf{1}$, where $p$ is a scalar defined in equation 8. Since the bias $b$ is a scalar, we use that $V_n\,b\mathbf{1} = bV_n\,\mathbf{1} = b\mathbf{1}$. We now consider the left-hand side of equation 16: $V_n\,\boldsymbol{Y}' = V_n\,(\boldsymbol{Y} + b\mathbf{1}) = V_n\,B_n(\boldsymbol{S})\,\boldsymbol{X} + V_n\,b\mathbf{1} = V_n\,B_n(\boldsymbol{S})\,\boldsymbol{X} + b\mathbf{1}$. Plugging the equality equation 16 into the last equation, we complete the proof: $V_n\,B_n(\boldsymbol{S})\,\boldsymbol{X} + b\mathbf{1} = B_n(\boldsymbol{S})\,\boldsymbol{R}\boldsymbol{X} + b\mathbf{1}$. □

This result allows us to increase the capacity of the equivariant hypersphere by adding the learnable parameter $b \in \mathbb{R}$.

### 4.3 NORMALIZATION AND ADDITIONAL NON-LINEARITY

An important practical consideration in deep learning is feature normalization (Ioffe & Szegedy, 2015; Ba et al., 2016). We show how the activations of the equivariant hypersphere (equation 13) can be normalized maintaining the equivariance:

**Proposition 6.** *Let $Y \in \mathbb{R}^{n+1}$ be the $\mathrm{O}(n)$-equivariant output of the hypersphere filter (equation 13). Then $Y/\|Y\|$, where $\|Y\| \in \mathbb{R}$, is also $\mathrm{O}(n)$-equivariant.*

*Proof.* Let $Y' = Y/\|Y\|$. We need to show that equation 16 holds also in the case of the normalization. We start by rewriting the right-hand side of equation 16: $V_n\,Y' = \|Y\|^{-1}V_n\,Y = \|Y\|^{-1}V_n\,B_n(S)\,X$. We then use the original equality in equation 16 and rewrite the last equation: $\|Y\|^{-1}V_n\,B_n(S)\,X = \|Y\|^{-1}B_n(S)\,RX$, which completes the proof. $\qquad\square$

To increase the descriptive power of the proposed approach, we can add non-linearity to the normalization step, following Ruhe et al. (2023):

$$Y \mapsto \frac{Y}{\sigma(a)\,(\|Y\| - 1) + 1}, \tag{17}$$

where $a \in \mathbb{R}$ is a learnable scalar and $\sigma(\cdot)$ is the sigmoid function.

### 4.4 EXTRACTING DEEP EQUIVARIANT FEATURES

We might want to propagate the equivariant output of $\mathbf{F}_n$ (equation 13), $Y = B_n(S)\,X$, through spherical decision surfaces while maintaining the equivariance properties. One way to achieve it is by using $(n+1)$D spheres, i.e., $\mathbf{F}_{n+1}$, since the output $Y \in \mathbb{R}^{n+1}$. Thus, the results established in the previous section not only allow us to use the equivariant hyperspheres (equation 13) for $n$D inputs but also to cascade them in multiple layers, thus propagating equivariant representations by successively incrementing the feature space dimensionality with a unit step, i.e., $n\mathrm{D} \to (n+1)\mathrm{D}$.

Consider, for example, the point cloud patch $\mathcal{X} = \{\mathbf{x}\}_{i=1}^{N}$ consisting of the coordinates of $N$ points $\mathbf{x} \in \mathbb{R}^n$ as the input signal, which we can also consider as the $N \times n$ matrix $\mathbf{X}$. Given the equivariant neuron $\mathbf{F}_n(\,\cdot\,; S)$, a *cascaded* $n\mathrm{D} \to (n+1)\mathrm{D}$ feature extraction procedure using equivariant hyperspheres $\mathbf{F}_n(\,\cdot\,; S)$ for the given output dimensionality $d$ (with $d > n$) can be defined as follows (at the first step, $X \leftarrow \mathbf{x}$):

$$\begin{aligned}
X \in \mathbb{R}^n &\to \mathtt{embed}(\mathtt{normalize}(X + b)) \to \mathbf{F}_n(X; S) \to \mathtt{embed}(\mathtt{normalize}(X + b)) \\
&\to \mathbf{F}_{n+1}(X; S) \to \dots \to \mathbf{F}_d(X; S) \to \mathtt{normalize}(X + b) \to X \in \mathbb{R}^d \,,
\end{aligned} \tag{18}$$

where $\mathtt{embed}$ is the embedding according to equation 1, $\mathtt{normalize}$ is the optional activation normalization (see Proposition 6), and $b$ is an optional scalar bias.

**Proposition 7.** *Given that all operations involved in the procedure 18 are $\mathrm{O}(n)$-equivariant, its output will also be $\mathrm{O}(n)$-equivariant.*

The proof is given in Section C.

Thus, given $\mathcal{X}$ as input, the point-wise cascaded application with depth $d$ (equation 18) produces the equivariant feature $\mathcal{Y} = \{Y\}_{i=1}^{N}$, $Y \in \mathbb{R}^{n+d}$, which we can consider as the $N \times (n + d)$ matrix $\mathbf{Y}$.

In this case, we considered the width of each layer in equation 18 to be 1, i.e., one equivariant hypersphere. In practice, one can use multiple equivariant hyperspheres per layer, with various types of connectivity between the layers, which is chosen based on the task at hand.

### 4.5 MODELLING HIGHER-ORDER INTERACTIONS

The theoretical framework established thus far considers the interaction of *one* point and one spherical decision surface (copied to construct the regular $n$-simplex constellation for the equivariant neuron in equation 13). To increase the expressiveness of a model comprised of equivariant hyperspheres, we propose to consider the relation of *two* points and a sphere, inspired by the work of Li et al. (2001).

Namely, given the input $\mathbf{X} \in \mathbb{R}^{N \times n}$ and the corresponding extracted equivariant features $\mathbf{Y} \in \mathbb{R}^{N \times (n+d)}$, we propose to compute

$$\Delta = \mathbf{E} \odot \mathbf{Y}\,\mathbf{Y}^\top, \tag{19}$$

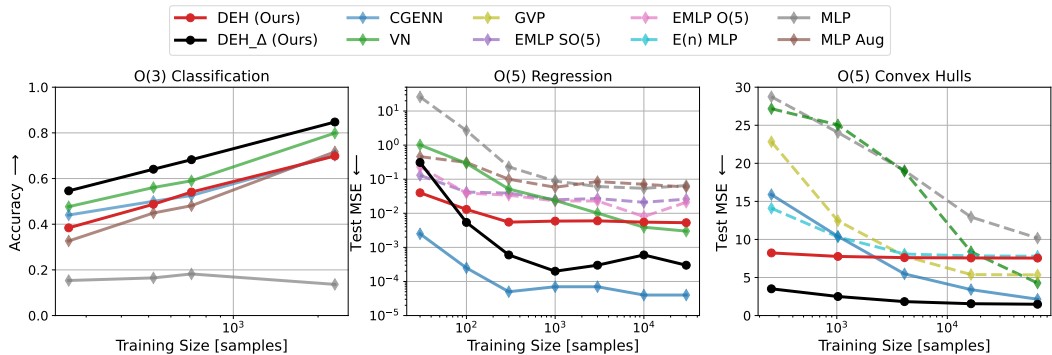

Figure 2: Left: real data experiment (the higher the accuracy the better); all the presented models are also permutation-invariant. Center and right: synthetic data experiments (the lower the mean squared error (MSE) the better); dotted lines mean that the results of the methods are copied from Ruhe et al. (2023) and the code for those particular versions of the models is unavailable. Best viewed in color.

where $\mathbf{E} := \frac{1}{2} \left( \|\mathbf{x}_i - \mathbf{x}_j\|^2 + \mathbf{I}_N \right) \in \mathbb{R}^{N \times N}$ models the edges as the squared distances between the points (with 1's in the main diagonal) in the input $\mathbf{X}$.

Note, that $\Delta \in \mathbb{R}^{N \times N}$ is O($n$)-invariant since $\mathbf{E}$ is comprised of the invariant distances between the points and the auto-product (Gram) matrix $\mathbf{Y}\mathbf{Y}^\top$ consists of the pair-wise inner products of equivariant features, which is invariant (Deng et al., 2021; Melnyk et al., 2022b). To enable permutation-invariance by aggregating over the points, we *first* follow the procedure by Xu et al. (2021) and sort the rows/columns of $\Delta$, and then apply max and/or mean pooling over $N$. If multiple ($K$) equivariant hyperspheres per layer are used, equation 19 is computed independently for each $K$, by broadcasting $\mathbf{E}$ and computing $K$ Gram matrices, resulting in $\Delta \in \mathbb{R}^{N \times N \times K}$.

## 5 EXPERIMENTAL VALIDATION

In this section, we experimentally verify our theoretical results derived in Section 4 by evaluating our `Deep Equivariant Hyperspheres`, constituting feed-forward point-wise architectures, on real and synthetic O($n$)-equivariant benchmarks. A more detailed description of the used architectures is presented in Table 1 in the Appendix. In addition to the performance comparison in Figure 2, we compare the time complexity (i.e., the inference speed) of the considered methods[4] in Figure 3.

### 5.1 O(3): ACTION RECOGNITION

First, we test the ability of our method to utilize O(3)-equivariance as the inductive bias. For this, we select the task of classifying the 3D skeleton data, presented and extracted by Melnyk et al. (2022a) from the UTKinect-Action3D dataset by Xia et al. (2012). Each skeleton is a $20 \times 3$ point cloud, belonging to one of the 10 action categories; refer to the work of Melnyk et al. (2022a) for details. We formulate the task to be both permutation- and O(3)-invariant.

We construct O(3)-equivariant point-wise feedforward models using layers with our equivariant hyperspheres (according to the blueprint of equation 18) with and without the two-point interaction described in Section 4.5, which we call DEH_$\Delta$ and DEH (see respectively the bottom and the top illustration in Figure D in the Appendix). We also build point-wise equivariant VN (Deng et al., 2021) and CGENN (Ruhe et al., 2023) models and non-equivariant baselines (MLPs, in which the equivariant layers are substituted with non-linear layers), all with the roughly the same number of the learnable parameters. We train the methods using the same hyperparameters. We train one version of the baseline with O(3)-augmentation, whereas our method is only trained on non-transformed skeletons. We evaluate the performance of the methods on the randomly O(3)-transformed test data.

---

[4]Some of the results are copied from Ruhe et al. (2023), and the implementation of the specific versions of some models is currently unavailable.

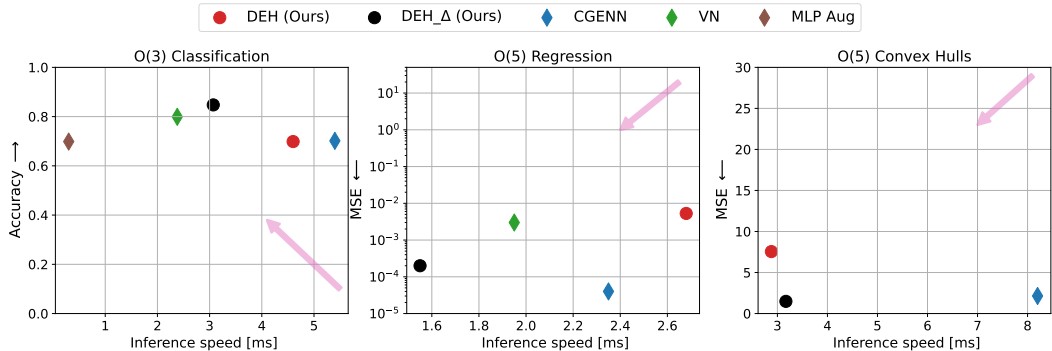

Figure 3: Speed/performance trade-off: The arrows in the plots articulate the direction of the desired trade-off, i.e., higher performance and faster inference. Best viewed in color.

The results are presented in Figure 2 (left): our models (both, DEH and DEH_Δ) trained on the data in a single orientation capture equivariant features sufficient to perform more effectively than the non-equivariant baseline trained on the augmented data (MLP Aug). Moreover, DEH_Δ consistently outperforms the competing equivariant methods (VN and CGENN), demonstrating a better speed/performance trade-off, as seen in Figure 3 (left).

## 5.2 O(5): REGRESSION

Originally introduced by Finzi et al. (2021), the task is to model the O(5)-invariant function $f(\mathbf{x}_1, \mathbf{x}_2) := \sin(\|\mathbf{x}_1\|) - \|\mathbf{x}_2\|^3/2 + \frac{\mathbf{x}_1^\top \mathbf{x}_2}{\|\mathbf{x}_1\|\|\mathbf{x}_2\|}$, where the two vectors $\mathbf{x}_1 \in \mathbb{R}^5$ and $\mathbf{x}_2 \in \mathbb{R}^5$ are sampled from a standard Gaussian distribution to construct train, validation, and test sets. We use the same training hyperparameters and evaluation setup as Ruhe et al. (2023). Here, we employ two models with architectures similar to those in Section 5.1, and compare them to the equivariant EMLPs (Finzi et al., 2021), CGENN, and VN, and non-equivariant MLPs.

Our results together with those of the related methods are presented in Figure 2 (center). As we can see, our DEH model exhibits fast convergence in terms of the training set size, and DEH_Δ outperforms the vanilla MLP and the MLP trained with augmentation (MLP Aug), as well as the O(5)- and SO(5)-equivariant EMLP (Finzi et al., 2021) and VN. Only CGENN outperforms our models, which comes, however, at the price of almost the double inference speed of our DEH_Δ (see the center of Figure 3).

## 5.3 O(5): CONVEX HULL VOLUME PREDICTION

We also consider the more challenging task of estimating the volume of the convex hull generated by 16 5D points, described by Ruhe et al. (2023). The problem is O(5)-invariant in nature. We exploit the same network architecture as in Section 5.1. As previously, we use the original training hyperparameters and evaluation setup presented by Ruhe et al. (2023).

We present our results alongside those of the related methods in Figure 2: Even our simplistic DEH model outperforms all the methods (including CGENN, GVP (Jing et al., 2021), and VN (Deng et al., 2021)) in the low-data regime (256 and 1000 training samples), and the DEH_Δ outperforms all of the competing methods in all the scenarios, exhibiting a superior speed/performance trade-off, as seen in Figure 3 (left).

## 6 CONCLUSION

In this manuscript, we presented `Deep Equivariant Hyperspheres` — $n$D neurons based on spheres and regular $n$-simplexes — equivariant under orthogonal transformations of dimension $n$. We defined and analyzed generalized components for a network composed of the proposed neurons, such as equivariant bias, non-linearity, and multi-layer configuration. We evaluated our method on both synthetic and real-world data and demonstrated the utility of the developed theoretical framework in $n$D, and achieved a particularly better trade-off in higher dimensions, as the O(5) experiments show a much clearer picture in Figure 3, Investigating the design of more advanced architectures of the proposed equivariant hyperspheres forms a clear direction for future work.

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

## A    ADDITIONAL BACKGROUND

### A.1    STEERABILITY

According to Freeman et al. (1991), a function is called *steerable* if it can be written as a linear combination of rotated versions of itself, as also alternatively presented by Knutsson et al. (1992). In 3D, $f^{\boldsymbol{R}}(x, y, z)$ is thus said to steer if

$$f^{\boldsymbol{R}}(x, y, z) = \sum_{j=1}^{M} v_j(\boldsymbol{R}) f^{\boldsymbol{R}_j}(x, y, z) , \qquad (20)$$

where $f^{\boldsymbol{R}}(x, y, z)$ is $f(x, y, z)$ rotated by $\boldsymbol{R} \in \mathrm{SO}(3)$, and each $\boldsymbol{R}_j \in \mathrm{SO}(3)$ orients the corresponding $j$th basis function.

Freeman et al. (1991) further describe the conditions under which the 3D steerability constraint (equation 20) holds and how to find the minimum number of basis functions, that must be uniformly distributed in space.

In this context, Melnyk et al. (2022a) showed that in order to steer a spherical neuron defined in equation 2 (Perwass et al., 2003; Melnyk et al., 2021), one needs to have a minimum of fours basis functions, i.e., rotated versions of the original spherical neuron. This, together with the condition of the uniform distribution of the basis functions, leads to the regular tetrahedron construction of the steerable 3D spherical neuron in equation 5.

## B    NUMERIC INSTANCES FOR $n = \{2, 3, 4\}$

To facilitate the reader's understanding of the algebraic manipulations in the next section, herein, we present numeric instances of the central components of our theory defined in equation 8 and equation 9, for the cases $n = 2$, $n = 3$, and $n = 4$. For convenience, we write the vertices of the regular simplex equation 8 as the $n \times (n + 1)$ matrix $\mathbf{P}_n = \left[\mathbf{p}_i\right]_{i=1\ldots n+1}$.

$$n = 2: \quad \mathbf{P}_2 = \frac{\sqrt{2}}{2} \begin{bmatrix} 1 & (\sqrt{3} - 1)/2 & -(\sqrt{3} + 1)/2 \\ 1 & -(\sqrt{3} + 1)/2 & (\sqrt{3} - 1)/2 \end{bmatrix}, \qquad p = \sqrt{3/2},$$

$$\mathbf{M}_2 = \frac{1}{\sqrt{3}} \begin{bmatrix} 1 & (\sqrt{3} - 1)/2 & -(\sqrt{3} + 1)/2 \\ 1 & -(\sqrt{3} + 1)/2 & (\sqrt{3} - 1)/2 \\ 1 & 1 & 1 \end{bmatrix}.$$

$$n = 3: \quad \mathbf{P}_3 = \frac{1}{\sqrt{3}} \begin{bmatrix} 1 & 1 & -1 & -1 \\ 1 & -1 & 1 & -1 \\ 1 & -1 & -1 & 1 \end{bmatrix}, \qquad p = 2/\sqrt{3},$$

$$\mathbf{M}_3 = \frac{1}{2} \begin{bmatrix} 1 & 1 & -1 & -1 \\ 1 & -1 & 1 & -1 \\ 1 & -1 & -1 & 1 \\ 1 & 1 & 1 & 1 \end{bmatrix}.$$

$$n = 4: \quad \mathbf{P}_4 = \frac{1}{2} \begin{bmatrix} 1 & (3\sqrt{5} - 1)/4 & -(\sqrt{5} + 1)/4 & -(\sqrt{5} + 1)/4 & -(\sqrt{5} + 1)/4 \\ 1 & -(\sqrt{5} + 1)/4 & (3\sqrt{5} - 1)/4 & -(\sqrt{5} + 1)/4 & -(\sqrt{5} + 1)/4 \\ 1 & -(\sqrt{5} + 1)/4 & -(\sqrt{5} + 1)/4 & (3\sqrt{5} - 1)/4 & -(\sqrt{5} + 1)/4 \\ 1 & -(\sqrt{5} + 1)/4 & -(\sqrt{5} + 1)/4 & -(\sqrt{5} + 1)/4 & (3\sqrt{5} - 1)/4 \end{bmatrix},$$

$$p = \sqrt{5}/2,$$

$$\mathbf{M}_4 = \tfrac{1}{\sqrt{5}} \begin{bmatrix} 1 & (3\sqrt{5}-1)/4 & -(\sqrt{5}+1)/4 & -(\sqrt{5}+1)/4 & -(\sqrt{5}+1)/4 \\ 1 & -(\sqrt{5}+1)/4 & (3\sqrt{5}-1)/4 & -(\sqrt{5}+1)/4 & -(\sqrt{5}+1)/4 \\ 1 & -(\sqrt{5}+1)/4 & -(\sqrt{5}+1)/4 & (3\sqrt{5}-1)/4 & -(\sqrt{5}+1)/4 \\ 1 & -(\sqrt{5}+1)/4 & -(\sqrt{5}+1)/4 & -(\sqrt{5}+1)/4 & (3\sqrt{5}-1)/4 \\ 1 & 1 & 1 & 1 & 1 \end{bmatrix}.$$

## C   COMPLETE PROOFS

In this section, we provide complete proof of the propositions and theorems stated in the main paper.

**Theorem.** *(Restating Theorem 4:)The neuron* $\mathbf{F}_n(\,\cdot\,;\boldsymbol{S}) : \mathbb{R}^{n+2} \to \mathbb{R}^{n+1}$ *defined in equation 13 is* $\mathrm{O}(n)$*-equivariant.*

*Proof.* We need to show that equation 3 holds for $\mathbf{F}_n(\,\cdot\,;\boldsymbol{S})$.

We substitute equation 14 to the LHS and equation 13 to the RHS, and obtain

$$V_n \, B_n(\boldsymbol{S}) \, \boldsymbol{X} = B_n(\boldsymbol{S}) \, \boldsymbol{R} \boldsymbol{X} \,. \tag{21}$$

Keeping in mind that the $(n+1)$-th and $(n+2)$-th components, $s_{n+1}$ and $s_{n+2}$, of the sphere $\boldsymbol{S} \in \mathbb{R}^{n+2}$ with center $\mathbf{c}_0 \in \mathbb{R}^n$ (equation 1) are $\mathrm{O}(n)$-invariant, as well as our convention on writing the rotation matrices (see the last paragraph of Section 3.2), we rewrite the $(n+1) \times (n+2)$ matrix $B_n(\boldsymbol{S})$ using its definition (equation 13):

$$B_n(\boldsymbol{S}) = \left[ (\boldsymbol{R}_O^\top \boldsymbol{R}_{T_i} \boldsymbol{R}_O \, \boldsymbol{S})^\top \right]_{i=1\ldots n+1} = \left[ \mathbf{c}_0^\top \boldsymbol{R}_O^\top \boldsymbol{R}_{T_i}^\top \boldsymbol{R}_O \quad s_{n+1} \quad s_{n+2} \right]_{i=1\ldots n+1}. \tag{22}$$

By definition of the rotation $\boldsymbol{R}_O$ (equation 13), we have that $\boldsymbol{R}_O \, \mathbf{c}_0 = \|\mathbf{c}_0\| \mathbf{p}_1$, where $\mathbf{p}_1 \in \mathbb{R}^n$ is the first vertex of the regular simplex according to equation 8. Since $\boldsymbol{R}_{T_i}$ rotates $\mathbf{p}_1$ into $\mathbf{p}_i$, we obtain

$$\boldsymbol{R}_{T_i} \boldsymbol{R}_O \, \mathbf{c}_0 = \|\mathbf{c}_0\| \mathbf{p}_i \,, \quad 1 \leq i \leq n+1 \,. \tag{23}$$

Thus, we can write the RHS of equation 21 using the sphere definition equation 1 as

$$B_n(\boldsymbol{S}) \, \boldsymbol{R} \boldsymbol{X} = \left[ \|\mathbf{c}_0\| \mathbf{p}_i^\top \boldsymbol{R}_O \quad s_{n+1} \quad s_{n+2} \right]_{i=1\ldots n+1} \boldsymbol{R} \boldsymbol{X} = \left[ \|\mathbf{c}_0\| \, \mathbf{P}_n^\top \boldsymbol{R}_O \boldsymbol{R} \quad s_{n+1} \mathbf{1} \quad s_{n+2} \mathbf{1} \right] \boldsymbol{X}. \tag{24}$$

We now use the definition of $V_n$ from equation 14 along with equation 10, equation 11, and equation 23 to rewrite the LHS of equation 21 as

$$\begin{aligned} V_n \, B_n(\boldsymbol{S})\boldsymbol{X} &= \mathbf{M}_n^\top \boldsymbol{R}_O \boldsymbol{R} \, \boldsymbol{R}_O^\top \mathbf{M}_n \left[ \left[ \|\mathbf{c}_0\| \, \mathbf{P}_n^\top \quad s_{n+1} \mathbf{1} \right] \boldsymbol{R}_O \quad s_{n+2} \mathbf{1} \right] \boldsymbol{X} \\ &= \mathbf{M}_n^\top \boldsymbol{R}_O \boldsymbol{R} \, \boldsymbol{R}_O^\top \left[ \left[ p \, \|\mathbf{c}_0\| \begin{bmatrix} \mathbf{I}_n \\ \mathbf{0}^\top \end{bmatrix} \quad \begin{matrix} \mathbf{0} \\ p\sqrt{n}\,s_{n+1} \end{matrix} \right] \boldsymbol{R}_O \quad \begin{matrix} \mathbf{0} \\ p\sqrt{n}\,s_{n+2} \end{matrix} \right] \boldsymbol{X} \\ &= \mathbf{M}_n^\top \boldsymbol{R}_O \boldsymbol{R} \left[ \begin{bmatrix} p \, \|\mathbf{c}_0\| & \mathbf{0} \\ \mathbf{0}^\top & p\sqrt{n}\,s_{n+1} \end{bmatrix} \boldsymbol{R}_O^\top \boldsymbol{R}_O \quad \begin{matrix} \mathbf{0} \\ p\sqrt{n}\,s_{n+2} \end{matrix} \right] \boldsymbol{X} \\ &= \frac{1}{p} \left[ \mathbf{P}_n^\top \boldsymbol{R}_O \boldsymbol{R} \quad n^{-1/2} \mathbf{1} \right] \begin{bmatrix} p \, \|\mathbf{c}_0\| \, \mathbf{I}_n & \mathbf{0} & \mathbf{0} \\ \mathbf{0}^\top & p\sqrt{n}\,s_{n+1} & p\sqrt{n}\,s_{n+2} \end{bmatrix} \boldsymbol{X} \\ &= \left[ \|\mathbf{c}_0\| \, \mathbf{P}_n^\top \boldsymbol{R}_O \boldsymbol{R} \quad \tfrac{\sqrt{n}}{\sqrt{n}}\,s_{n+1} \mathbf{1} \quad \tfrac{\sqrt{n}}{\sqrt{n}}\,s_{n+2} \mathbf{1} \right] \boldsymbol{X} \quad = \quad B_n(\boldsymbol{S}) \boldsymbol{R} \boldsymbol{X}. \end{aligned} \tag{25}$$

$\square$

**Proposition 8.** *(Restating Proposition 7:) Given that all operations involved in the procedure 18 are* $\mathrm{O}(n)$*-equivariant, its output will also be* $\mathrm{O}(n)$*-equivariant.*

*Proof.* Let $\mathbf{R} \in \mathrm{O}(n)$ be an orthogonal transformation, $\rho_i(\boldsymbol{R})$ the representation of $\boldsymbol{R}$ in the respective space, e.g., equation 14 for the equivariant hypersphere output, and $\boldsymbol{x} \in \mathbb{R}^n$ be the input to the procedure 18. We denote the output of the procedure 18 as $\mathbf{F}(\mathbf{x})$, where $\mathbf{F}$ is the composition of all operations in the procedure 18. Since each operation is equivariant, equation 3 holds for each operation $\Phi$, i.e., we have $\Phi_i(\rho_i(\boldsymbol{R})X) = \rho_{i+1}(\boldsymbol{R})\Phi(X)$. Consider now the output $\mathbf{F}(\mathbf{x})$ and the transformed output $\mathbf{F}(\boldsymbol{R}\mathbf{x})$. Since each operation in $\mathbf{F}$ is equivariant, we have: $\mathbf{F}(\boldsymbol{R}\mathbf{x}) = \Phi_d(\Phi_{d-1}(\dots\Phi_2(\Phi_1(\boldsymbol{R}\mathbf{x})))) = \rho_d(\boldsymbol{R})\Phi_d(\Phi_{d-1}(\dots\Phi_2(\Phi_1(\mathbf{x})))) = \rho_d(\boldsymbol{R})\mathbf{F}(\mathbf{x})$. Thus, the output of the procedure in equation 18 is equivariant, as desired. $\square$

## D ARCHITECTURE DETAILS

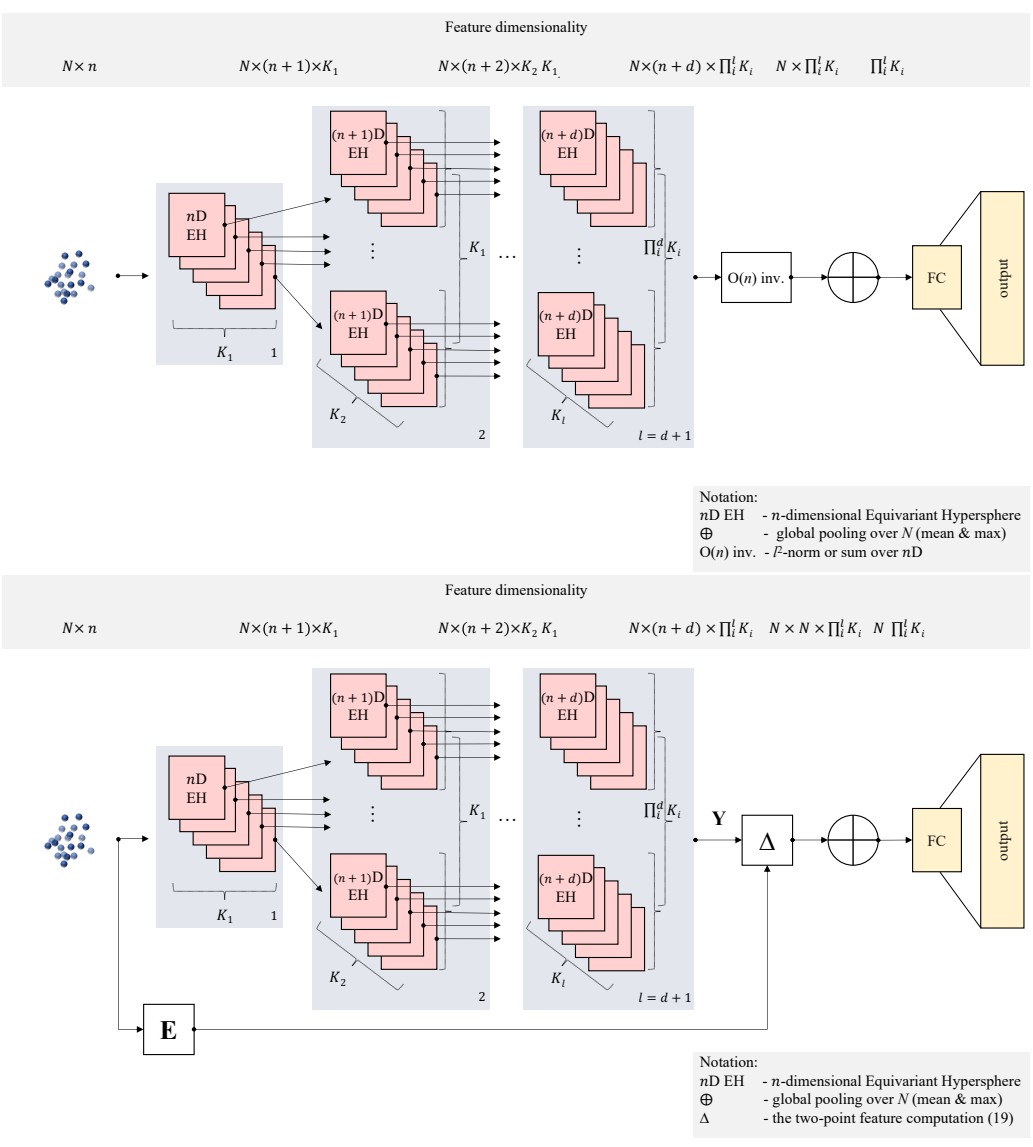

Figure 4: Architectures of our model: DEH (top) and DEH_$\Delta$ (bottom). All the operations are pointwise, i.e., shared amongst $N$ points. Each subsequent layer of equivariant hyperspheres contains $K_l$ neurons for each of the $\prod_i^d K_i$ preceding layer channels. The architectures of the non-permutation-invariant variants differ only in that the global aggregation function over $N$ is substituted with the flattening of the feature map.

| Methods | Equivariant layer sizes | Invariant operation | FC-layer sizes | Total #params |
|---------|------------------------|--------------------|----------------|---------------|
| | | O(3) Action Recognition | | |
| DEH | [8, 48, 96] | sum | 32 | 7.81K |
| DEH_$\Delta$ | [3, 6] | $\Delta$ | 32 | 8.11K |
| | | O(5) Regression | | |
| DEH | [2, 2] | $l^2$-norm | 32 | 391 |
| DEH_$\Delta$ | [2] | $\Delta$ | 32 | 343 |
| | | O(5) Convex Hulls | | |
| DEH | [32, 512] | $l^2$-norm | 32 | 38.2K |
| DEH_$\Delta$ | [8, 48] | $\Delta$ | 32 | 49.8K |

Table 1: Model architectures and number of parameters used in the experiments.

In this section, we provide illustrations of the architectures of our models used in the experiments in Section 5 (see Figure D). By default in all our models (DEH and DEH_$\Delta$), we learned non-normalized hyperspheres and equipped the layers with the equivariant bias and the additional non-linearity (non-linear normalization in equation 17). The number of learnable parameters corresponds to the competing methods in the experiments.

