# OpenReview forum: "Learning Deep O($n$)-Equivariant Hyperspheres"
_ICLR.cc/2024/Conference — Submitted to ICLR 2024_

### Official Review · Reviewer_gZ3c · 2023-10-31

**Soundness:** 3 good
**Presentation:** 3 good
**Contribution:** 2 fair
**Rating:** 6
**Confidence:** 4

**Summary:**

The paper demonstrates how to build neural networks with cascaded layers of spherical neurons (neurons with spherical decision surfaces) that are equivariant to the action of the orthogonal group $O(n)$. Earlier works have only considered $O(3)$-equivariant spherical neurons in the first layer (i.e., no cascaded spherical neuron layers). The paper also describes how to layers with bias and normalization and proves exact equivariance to $O(n)$ action in all cases.

**Strengths:**

1. The paper is easy to follow and the ideas are developed and presented well.

2. The proposed architecture seems to produce good results for some tasks tested on compared to strong baselines including Vector Neurons and EMLPs in many cases.

**Weaknesses:**

1. One of the weaknesses is the motivation for this architecture. I am not able to see why spherical neurons, their equivariant versions, and cascaded layers of such neurons are an improvement over conventional multi-layer perceptrons (MLPs) and Equivariant MLPs (EMLPs), other than looking at experimental results.

2. Although the architecture is technically novel compared to the earlier work on equivariant spherical neurons in 3D for the single layer case, the generalization appears to be straightforward, the arguments follow just by increasing the dimension from 3 to $n$.

3. The limitation of the architecture is quite large. In the case of the convex hull prediction task, many algorithms outperform the proposed one when the training set is large. The authors say this is because higher-order interactions are not present in this architecture. Both VN and GVP outperform the proposed algorithm, but do not model second-order interactions. Furthermore, VN and GVP are tested only for the convex hull prediction task, not the other two tasks, so this leaves some questions about how good this architecture actually is.

**Questions:**

No additional questions, but it would be good if the authors can address the weaknesses I mentioned.

---

> ### Author Response · Authors · 2023-11-15
>
> Thank you for your review and assessment of our work!
>
>
> Below we address the Weaknesses you pointed out:
>
>
> 1. - Our intuition as to why spherical neurons and equivariant hyperspheres in $n$D offer advantageous modeling is as follows: the relation between a point and a spherical decision surface (as the generalization of planar decision surfaces represented by the vanilla MLPs) is isometric (Melnyk et al., 2021), which allows for an O($3$)-equivariant construction of the steerable spherical neuron (Melnyk et al., 2022a), preserving the geometry of the input space. This was previously demonstrated only for $n=3$ and only for the input layer (Melnyk et al., 2022a,b).
>
>
>
>     - In our manuscript, we extrapolate these ideas: Melnyk et al. 2022a show that the output of an O(3) layer is a subgroup of O(4), therefore again an isometry.
>
>     - Thus, similar to the argument in Melnyk et al. 2021, a 4D sphere is more appropriate than hyperplanes. Setting an O(4)-equivariant layer on top produces an isometric representation in 5D. We show that we can do this in arbitrary dimensions.
>
>
> 2. Indeed, the proposed neurons are the generalization of the 3D neurons. The *correctness* of the generalization to $n$D, however, is non-trivial and requires the rigorous proof that we provide in the manuscript (Theorem 4).
>
> 3. - We focused on the single point case to verify our hypothesis, which we still believe is confirmed. This verification is necessary for any higher-order extension. If the absolute level of performance is considered to be the most important aspect, we can include our suggested two-point version (and add all the other details). For this, please see the “Low performance” part of the general comment as well as the results (the updated Figure 2 in the manuscript) for the two-point version of our model, showing the superiority of our method over all the models, including both VN and GVP, in the O(5) Convex Hulls task, as well as in the O(3) Action recognition task, including CGENN.
>
>     - In relation to competing work, the experimental setup and results for O(5) Regression and O(5) Convex Hulls are used as-is from Ruhe et al. (2023), as we stated in Sections 4.2 and 4.3, as well as the general comment. We do not have access to the code of the specific models used in those experiments other than those provided in https://github.com/DavidRuhe/clifford-group-equivariant-neural-networks. We will, however, provide all the code for the experiments and models that we had access to and ran ourselves.
>
>     - If the time before the rebuttal deadline allows and it is suggested to be necessary, we will try to conduct additional experiments for some of the competing models, which we will need to create ourselves (since no code is available for further experiments except for those we already included).
>
>   **UPD**: We have now conducted additional experiments with VN. We will include them and update the manuscript before the rebuttal deadline.
>
> Please do let us know if there is anything we need to clarify further.

---

> > ### Comment · Reviewer_gZ3c · 2023-11-22
> > **Thank you for the response**
> >
> > Thank you for the response.
> >
> > I am still not very convinced on the theoretical novelty going from three dimensions to an arbitrary dimension. I also don't see the clear advantage of isometry between the input space and the feature space.
> >
> > I do appreciate the additional experimental results that clarify some of the contributions and increase my rating by one level.

---

> > > ### Author Response · Authors · 2023-11-23
> > >
> > > We thank the reviewer for the response and highly appreciate the raised rating!
> > >
> > > We have now updated our manuscript, in which we implemented all the suggested changes and added the results.

---

### Official Review · Reviewer_sbi9 · 2023-11-02

**Soundness:** 3 good
**Presentation:** 2 fair
**Contribution:** 2 fair
**Rating:** 6
**Confidence:** 2

**Summary:**

This paper introduces a novel Deep Equivariant Hypersphere model that naturally exhibits group equivariance and invariance properties for arbitrary O(n) groups encompassing rotation and reflection operations in n-dimensional spaces. This achievement is made possible through the utilization of spherical neurons and the generalized steerable function theorem. While prior research predominantly concentrated on symmetry within O(3), which involves rotation and reflection operations in three-dimensional space, this paper distinguishes itself by extending the concept to a broader context, the O(n) groups. The authors delve into various theoretical aspects of the proposed model and substantiate its capabilities through experiments, showcasing its effectiveness in learning O(n) equivariance and excelling in tasks such as O(n) invariance regression and classification.

***

[POST-REBUTTAL] Thank you for clarifying my confusion regarding the working principle of the proposed model, and addressing my question about generalization to other groups. I also appreciate the inclusion of Figure 4 and additional details on the model description and hyper-parameters in the revised paper. Considering these, I would like to maintain my score.

**Strengths:**

In the field of geometric deep learning, it is essential to develop a model that can effectively capture O(n) symmetry. This paper achieves this goal by utilizing well-defined geometric techniques, specifically spherical neural networks and steerable functions.

**Weaknesses:**

I have doubts about whether the model proposed in this paper can be easily applied to real-world mixed-transformed datasets.

If I understand correctly, the overall proposed approach can be summarized as follows: First, a spherical neuron-based model is created to match the dimensionality of the given dataset. Then, n + 1 transformed copies of this baseline model are generated through applying O(n) transformations (using a form of n-simplex). These transformed n + 1 models serve as the basis functions for constructing steerable functions, resulting in the creation of an O(n) equivariance model called the Deep Equivarint Hyperspheres.

Considering the definition of steerable functions, these generated bases represent rotations of the original function in specific directions, implying that each copy of the spherical neuron model should have learned data aligned with a single specific orientation. In other words, to create a Deep Equivariant Hypersphere, it is essential to have a dataset aligned with a canonical frame, to initially train the baseline spherical neurons. Indeed, the experiments in this paper demonstrate that a model trained on data aligned with the canonical orientation can effectively generalize to randomly transformed test datasets. This experiment is certainly meaningful.

However, in real-world scenarios, we often need to learn equivariance or invariance for datasets that are already randomly transformed. In most cases, we cannot know the angles or group actions applied to instances within a given dataset beforehand. Obtaining a dataset aligned with the canonical orientation is often challenging in practice. I believe that in such situations, the proposed Deep Equivariant Hypersphere may not perform effectively.

Another concern with this paper is that, while the paper rigorously explains the theoretical properties of the proposed model, it seems to lack sufficient explanation of how to construct a practical model and training algorithm in practice. At the very least, I suggest that such information should be added to the appendix. The current version of the paper is quite challenging to follow, especially for those who are not familiar with the two prior papers authored by Melynk et al [1, 2].

***

[1] Melnyk, P., Felsberg, M., & Wadenbäck, M. (2021). Embed me if you can: A geometric perceptron. In Proceedings of the IEEE/CVF International Conference on Computer Vision (pp. 1276-1284).

[2] Melnyk, P., Felsberg, M., & Wadenbäck, M. (2022, June). Steerable 3D spherical neurons. In International Conference on Machine Learning (pp. 15330-15339). PMLR.

**Questions:**

The paper primarily emphasizes O(n) groups and acknowledges their importance in real-world scenarios. However, within the realm of geometric deep learning literature, a significant challenge lies in developing models that can generalize effectively for broader Lie groups beyond O(n). I am curious to know if the proposed Deep Equivarint Hyperspheres can effectively learn and handle such general Lie groups.  I understand that concepts such as the spherical neuron or n-simplex used in this paper may not easily apply to other Lie groups. However, I would like to hear the authors' thoughts on the extension of steerable functions for arbitrary Lie groups.

---

> ### Author Response · Authors · 2023-11-14
>
> First of all, thank you for your positive review and assessment!
>
> Your summary accurately lists our main contributions.
>
> _______
> We begin by addressing the Weaknesses you pointed out:
>
> - We learn a model consisting of deep equivariant hyperspheres $B(S)$ defined in equation (13): that is, each $B(S)$ is a matrix $ \in \mathbb{R}^{(n+1)\times d}$ and has only one row $S \in \mathbb{R}^d$ as a learnable parameter vector – the rest adhere to the n-simplex construction.
> Therefore, we don’t need to copy the trainable spherical decision surfaces after training – it is already encapsulated in $B(S)$.
>
> - It is not necessary to have training data aligned:
> (For the 3D case, it’s been demonstrated in “TetraSphere” by Melnyk et al. (2022), in which the first layer consists of 3D equivariant neurons $B(S)$.)
>     - To demonstrate this in our case, we conducted the Action recognition experiment, in which we both trained the model on the transformed (“O(3)”) training data and tested it on the transformed test data (“O(3)”, as before), resulting in an O(3)/O(3) train/test augmentation setup (as compared to no-rotation augmentation during training, “I”, in the original experiment, I/O(3)).
>
>     -	Here are the test accuracies of our DEH_PI model (for the maximum number of training data samples):
>
> |I/I (for reference)   |     I/O(3) (the original)     |     O(3)/O(3)|
> |-------------|------------- |------------- |
> |69.9%        |                                  69.9%      |                                    69.8% |
>
>
>
> - We train all the models end-to-end, using Adam optimizer.
> In this regard, our model is no different from the vanilla MLP or any other model in the experimental comparison presented in the paper.
>
>
> _______
> Addressing your Questions:
>
> - In general, the main limitation is the required compactness:
>     - Our concept can be generalized, but with a finite number of basis functions, we can only cover a compact group (or compact interval of a non-compact group). O(n) (presumably even U(n)) is a very powerful group that covers many other groups as subgroups.
>
>
> _______
> We sincerely hope that this clarifies our method.
> Please let us know should you have additional questions.

---

### Official Review · Reviewer_MJcE · 2023-11-03

**Soundness:** 4 excellent
**Presentation:** 1 poor
**Contribution:** 2 fair
**Rating:** 5
**Confidence:** 3

**Summary:**

This paper generalizes spherical neurons to the orthogonal group of dimension greater than 3, yielding a scalable $O(n)$-equivariant architecture. Their architecture creates steerable filter banks based on the higher-dimensional simplex. They evaluate their architecture on an $O(3)$-invariant classification dataset, as well as on two synthetic $O(5)$-equivariant datasets, and obtain strong results in the low sample complexity regime.

**Strengths:**

The extension of spherical neurons to higher dimensions is novel, and this work presents an $O(n)$-equivariant architecture that is distinct from previous $O(n)$-equivariant architectures and requires nontrivial theoretical development. Their experimental results are better than their selected baselines in low sample regimes. The proofs are clearly written and thorough.

**Weaknesses:**

1. Generally, there is a severe lack of comparison to other O(n)-invariant and -equivariant architectures, such as EGNN, EMLP, vector neurons, canonicalization (alt. frame-averaging), and for invariance, simply taking inner products. O(n) equivariance is not new, so what is superior about this particular framework? If the benefit over baselines like CGENN is inference speed, this should be recorded in a table or figure. Many of these baselines were not included in the experiments, either, especially the only non-synthetic experiment (O(3)-invariant skeleton classification). It would also be helpful to articulate a theoretical comparison between the hyperspheres approach and existing approaches — which is more expressive, which has better scaling, which (if any) can be considered a special case of hyperspheres or vice versa.
2. The presentation of the paper could be significantly improved. Not having seen spherical neurons or TetraSphere before, I found the paper’s writing and architecture description very hard to follow — even after checking these original papers. I remain unsure what the guiding intuition for the hyperspheres equivariant framework is. Also, I would suggest making this paper more self-contained in a future revision. For example, steerability in section 2.3 is a crucial concept that is defined only with high-level words, whereas an equation would be important to properly understand the concept (especially since it is so overloaded). The subsequent claim that a “3D steerable filter consisting of spherical neurons needs to comprise a minimum of four 3D spheres” is also unclear; the background on spherical neurons in section 2.1 does not provide enough detail to resolve the meaning of this sentence without checking the original paper. Moreover, the proofs in the main body of the paper could be moved to the appendix, to make space for figure and writing that better clarifies the method and appeal of hyperspheres. For example, Proposition 7 simply states that composing equivariant layers yields and equivariant end-to-end function; this is quite standard and intuitive, and probably does not merit a full paragraph’s explanation in the main body.
3. The only experimental comparisons to a real dataset is on an O(3)-invariant action recognition dataset. However, the main innovation of the paper is for O(n) with n>3. Justification of the practical applications of O(n)-equivariance would much better motivate this work.
4. Even on the simulated O(5) regression datasets, performance is worse than the CGENN baseline.

**Questions:**

1. How expressive is the proposed $O(n)$-equivariant architecture? Is it universal, i.e. can it represent any continuous $O(n)$-equivariant function?
2. As the authors find suboptimal experimental performance relative to methods like CGENN which have higher order interactions between the points, is it possible that there’s an expressivity gap between equivariant hyperspheres and methods which enjoy higher order equivariance?
3. How well did each method perform without any test-set rotation augmentation on the real dataset (UTKinect-Action3D)?
4. Why is there no comparison to equivariant baselines on the real dataset (UTKinect-Action3D)? Moreover, why is there no comparison to equivariant methods other than EMLP, such as vector neurons, EGNN, etc?
5. How are equivariant hyperspheres related to the irreducible representations of $O(n)$, if at all?
6. What are some practical applications of $O(n)$-equivariance, for $n>3$?

---

> ### Author Response · Authors · 2023-11-17
>
> Thank you very much for your detailed feedback and assessment of our work!
> We appreciate that our contributions are accurately identified.
>
> We first address the Weaknesses:
>  _____
> W1.
> - Why our framework is superior:
>
>      - as we state in the last paragraph on p. 1, *“Similarly to the approach of Ruhe et al. (2023), Deep Equivariant Hyperspheres directly operate on the basis of the input points, not requiring constructing an alternative one, ..., which is a key limitation of many related methods (Anderson et al., 2019; Thomas et al., 2018; Fuchs et al., 2020).”*, as well as the
>
>      - inference speed: below we provide a comparison of the inference time of the models in the O(5) Convex Hulls task (the models available to us), for which we used the NVIDIA A40:
>
> |DEH | DEH_PI | DEH_$\Delta$_PI | CGENN |
> |-------------|-------------|-------------|-------------|
> | 4.3 ms | 2.7 ms | 3.1 ms | 8.2 ms |
>
> , where DEHs are our models (see the updated Figure 2).
>
> - Including baselines in the experiments:
>
>     - O(5) Regession and O(5) Convex hulls: the experimental setup and results are used as-is from Ruhe et al. (2023), as we stated in Sections 4.2 and 4.3, as well as the general comment. We do not have access to the code of the specific models used in those experiments other than those provided in https://github.com/DavidRuhe/clifford-group-equivariant-neural-networks. We will, however, provide all the code for the experiments and models that we had access to and ran ourselves. **UPD** We have conducted additional VN experiments, and will include them in the manuscript before the rebuttal deadline.
>
>     - O(3) Action recognition: we have now included a permutation-invariant version of CGENN (please see the updated Figure 2).
>
> - Theoretical comparison: in addition to the theoretical comparison presented in the last paragraph on p. 1 and the first paragraph on p. 2, we add that  the EMLP is a representation method building equivariant feature maps by computing an integral over the respective group (which is intractable for continuous Lie groups and hence, requires coarse approximation); the E(n)-MLP version of EGNN in the comparison is similar to our method in that it operates on scalars: it updates the vector information by learning the parameters conditioned on scalar information and multiplying the vectors with it;  the vector neurons, in contrast to our equivariant scheme, operate on latent features of each point in an equivariant manner, and the invariant features are obtained by taking the scalar product of the latent features per point; the vanilla MLP representing planar decision surfaces is, in fact, a particular case of spherical decision surfaces (Perwass et al., 2003; Melnyk et al., 2021), and thus, can be seen as a particular case of the proposed hyperspheres.
>
> W2.
> -  In general, the presented model enjoys PointNet-like point-wise feature extraction (but with our equivariant hyperspheres).
> In addition to the architecture description in the second paragraph in Section 4.1, we will provide (**UPD** the illustrations in the updated manuscript before the rebuttal deadline) and the code, which should facilitate the reader’s understanding.
> - We appreciate the suggestion of making our paper more self-contained. We will add more background to it and adjust the relative sections accordingly.
> - The guiding intuition behind the hypersphere framework is as follows:
>          - the relation between a point and a spherical decision surface (as the generalization of planar decision surfaces represented by the vanilla MLPs) is isometric (Melnyk et al., 2021), which allows for an O(3)-equivariant construction of the steerable spherical neuron (Melnyk et al., 2022a), preserving the geometry of the input space;
>          - we extrapolate these ideas: Melnyk et al. 2022a shows that the output of an O(3) layer is a subgroup of O(4), therefore again an isometry. Thus, similar to the argument in Melnyk et al. 2021, a 4D sphere is more appropriate than hyperplanes. Setting an O(4)-equivariant layer on top produces an isometric representation in 5D. We show that we can do it in arbitrary dimensions.
>
> W3. Experiments and practical applications for $n>3$:
>
> - the choice of the experiments for $n>3$ is motivated by the related work (Finzi et al., 2021; Ruhe et al., 2023);
>
> - a potential practical application for $n>3$ would be for the cases when 3D points (e.g., a point cloud) contain invariant features with $d$ dimensions (e.g., color or density information). Then one could consider the input signal to be $(n+d)$-dimensional and apply the O(n)-equivariant framework. However, there are no results or code for competing methods available for that case and the effort for re-writing the original code is too substantial to be done during the rebuttal.
>
> W4. Please see the “Low performance” part of our general comment and the updated Figure 2.
> ______________________________________

---

> ### Author Response · Authors · 2023-11-17
> **Addressing the Questions**
>
> Q2. We attribute the suboptimal performance not to the equivariant hyperspheres (i.e., the neurons in equation 13) themselves, but to the absence of higher-order interactions between the extracted equivariant features of the points:
> - As indicated by the performance of the two-point version of our model (which we will add with all the details if advised; see the “Low performance” part of our general comment and Figure 2), which maintains the original equivariant feature extraction scheme, our model outperforms the related methods in the O(3) Action recognition and O(5) Convex Hulls tasks.
>
> Q1. Given the time constraints, we can only provide an answer based on the experimental verification. The expressiveness of the original architecture is limited due to the absence of higher-order interaction modeling (see "Limitations" on p.9 and our answer to Q2). The two-point version of our model outperforms, among other models, GVP, which has been proven to be a universal function approximator for O(3) (Jing et al., 2021), and other expressive equivariant architectures like VN (Deng et al., 2021) and CGENN (Ruhe et al., 2023).
>
> Q3. O(3) Action recognition experiment without transforming test set (“I”) vs. with O(3)-transformation (“O(3)") for the maximum number of training samples:
>
> |Test aug.\Model| DEH_PI   |  DEH_$\Delta$_PI  |   CGENN_PI | MLP| MLP+Aug|
> |-----|:-------------:|:-------------: |:-------------: | :-------------:|  :-------------:|
> |I|   69.9% | 84.7%  |70.2% | **91.8%** | 72.6% |
> |O(3) | 69.9% |**84.7%** | 70.2% | 14.1% | 71.7% |
>
> , where all the models are permutation-invariant, and DEHs are our models; please see the general comment.
>
> (Since the models DEH_PI, DEH_$\Delta$_PI , and CGENN_PI produce O(n)-invariant predictions, the results are identical to those presented for randomly transformed test data in Figure 2 in our manuscript.)
>
> Q4. We have now added the equivariant CGENN, which initially outperformed all other models in the two O(5) tasks (please see Figure 2), to the experiments.
> If the time before the rebuttal deadline allows, we will try to add another.
>
> Q5. Relation to the irreducible representation of O$(n)$:
> - we appreciate the interesting question! ~We are looking into it and will respond shortly.~
> - UPD: Our $\textbf{M}$-$\textbf{\textit{R}}_{O}$ decomposition (14) appears to be unrelated to the Clebsch-Gordan decomposition, and hence, our method is not based on the obtained irreducible representations.
>
> Q6. Please see the second bullet in our answer to W3.
>
> ________
> We sincerely hope that our answers address the concerns.
> We are working on incorporating the suggested changes into the manuscript.
>
> Please let us know if we need to provide additional clarification.

---

> > ### Comment · Reviewer_MJcE · 2023-11-22
> > **Thank you for the rebuttal**
> >
> > Thanks to the authors for their detailed response. A few responses below:
> >
> > * Why is "directly operat[ing] on the basis of the input points" a limitation, independently of the forward pass time? I was under the impression this was the primary advantage.
> > * The inference speed results shown are encouraging! In a final version of the paper, it would be helpful to have a scatter plot of inference time vs test error, where each point is a method, to validate whether this method achieves a better speed/error tradeoff at any point on the plot. Other baselines should also be added to make this point.
> > * An articulation of the expressiveness of O(n)-equivariant hyperspheres, as compared to other baselines, is still absent. This seems to me an important facet to understand before a new method is published.
> > * The two-point version of the network inspired by Li et al indeed seems to work better, although this is a departure from the submitted work; an explanation of the two-point version in the revision is necessary.
> > * The proposed method is less accurate than an MLP on the original UTKinect-Action3D dataset (and this new comparison lacks VN, GVP etc).
> > * In the second bullet point response to W3, do the authors mean an $O(n+3)$-equivariant network, not an $O(n)$-equivariant network? And if so, if the $n$ features are invariant, this essentially sounds to me like $O(3)$ equivariance, but maybe I am misunderstanding.
> >
> > With these points in mind, I will retain my original rating. However, I encourage the authors to resubmit to a later venue, especially after having had a chance to perhaps add more extensive experiments (including the two-point version) with baselines and timing tests, a more thorough theoretical comparison to baselines in terms of expressivity, and hopefully a more self-contained presentation of the method.

---

> > > ### Author Response · Authors · 2023-11-23
> > >
> > > We thank the reviewer for the comments, suggestions, and encouragement!
> > > We provide additional clarification:
> > >
> > > - We are sorry for the confusing quote: indeed, the meaning of the quote is that *"requiring constructing an alternative [basis] ... is a key limitation of many related methods (Anderson et al., 2019; Thomas et al., 2018; Fuchs et al., 2020)"*. Similar to Ruhe et al. (2023), our method is advantageous in this sense, since it operates directly on the basis of the input points.
> > >
> > > - We have now added the suggested scatter plot for all the experiments and available models, as seen in Figure 3.
> > >
> > > - We would like to articulate the original motivation of the expressiveness of the equivariant hyperspheres with the following table
> > > |Method| Spherical decision surfaces | O(n)-equivariance|
> > > |----------|------------|--------------|
> > > |Vanilla MLP| no       | no              |
> > > | CGENN/EMLP/E(n)MLP/VN/GVP| no | yes |
> > > |Deep Equivariant Hyperspheres (Ours)| yes | yes|
> > >
> > > As stated in the Introduction, the utility of spherical decision surfaces (as well as the expressiveness, as compared to their planar counterpart) come from prior work (Perwass et al. (2003), Melnyk et al. (2021)).
> > > The motivation of O(n) equivariance is given in the Introduction, as well as the body of related work (now recapped in Section 2).
> > > Our method combines the expressiveness of the spherical decision surfaces and the utility of O(n)-equivariant feature extraction, thus fulfilling the last row in the Table above.
> > >
> > >
> > > - We have added a description of the two-point model in Section 4.5 (as well as Section D in the Appendix). As can be seen from the architecture illustrations (see Figure 4 in the Appendix), which we added as per our answer to W2, the two-point version (at the bottom) has only minimal change compared to the one-point model (at the top), and maintains the same equivariant hyperspheres structure.
> > >
> > > - We would like to point out that the purpose of the O(3) experiment in particular is to test the ability of the models to perform equally well when the test data is in unseen orientations (a desired property for point cloud classification). The performance of the MLP in this scenario drops from 91.8% to 14.1%, whereas our method maintains a high performance of 84.7%.
> > >
> > > - The reviewer is correct: the example we have given is still intrinsically O(3)-equivariant. Perhaps, a more proper practical higher-dimensional example would be having input as 3D points appended with the coordinates of the normal(s), thus forming 6D vectors. This task would be G-equivariant, where O(3)<G<O(6).
> > >
> > >
> > > ____________
> > > As per the reviewer's suggestions, we have updated the manuscript and added an additional background section in the Appendix, as well as the architecture illustrations (Figure 4), competing methods experiments (Figure 2), and time plots (Figure 3); please see the updated pdf.

---

### Official Review · Reviewer_RNHY · 2023-11-04

**Soundness:** 3 good
**Presentation:** 3 good
**Contribution:** 3 good
**Rating:** 6
**Confidence:** 5

**Summary:**

The paper presents Deep Equivariant Hyperspheres - a class of neural networks equivariant under $O(n)$ transformation group. The model is based on spherical neurons and regular $n$-simplexes. The authors present the idea of $O(n)$ equivariant features of a special type, then they study their properties when such features are composed one after another and form a deep model.

While the theory is correct and coherent, the experimental evaluation of the method is very limited. It does not allow a reader to infer the advantage of the proposed method.

After reading the revised version of the paper, and the answers of the authors, I am increasing the rating.

**Strengths:**

- The paper is well-written. It uses mathematical notation when it's required and the rest is clearly explained with plane text. It makes it easy to follow the ideas of the authors
- The presented method is alternative to more common MLP-based or graph-convolution-based models. It makes the field richer in terms of approaches.
- The theory is mathematically correct

**Weaknesses:**

The main weakness of the paper is that the authors do not clearly demonstrate the real-life fields where this method is most advantageous and what its limitations are. To address this, it makes sense to:

1. **Organize a proper related work section:** The current version uses the Introduction and Background to position the current paper in the field of previously developed methods. This approach is rather implicit and doesn't allow the reader to properly understand which papers served as inspiration for the authors, which ones are direct competitors, and which ones are not relevant at all but are mentioned.

2. **Discuss the limitations:** The authors can significantly improve the quality of the paper by explicitly discussing the limitations of the approach or even demonstrating them experimentally.

3. **Describe the hyperparameters:** The current paper lacks a description of the hyperparameters of the method. It would improve the quality if the authors described the main hyperparameters of the proposed approach and conducted ablation studies on them.

Another significant weakness is the **limited experimental evaluation** of the proposed method. The current experimental results demonstrate some improvement over a subset of the competitors. However, it is achieved on small datasets, with small models and only in the very-low-data regime. The common number of training samples in the experiments for which the proposed method outperforms the others is 1000. A datasets of 1000 samples is unrealistically small for the field of machine learning nowadays.

**Questions:**

- In $O(5)$ regression experiment, CGENN significantly outperforms your method. Is the use of higher order tensor features in CGENN the only reason it performs better? If so, can you train a modification of CGENN which uses only scalar and vector features?
- In $O(5)$ convex hulls experiment, your method perfoms worse than other, when the number of training samples increases. Can you demonstrate the effect with $10^6$ samples?
- In both $O(5)$ experiments the proposed method reaches a plateau. While some of the other methods continue to improve. I can't infer the advantage of the proposed method from the plots. All 3 plots should be extended to a larger number of samples, because so far it looks like you demonstrated only a small part of the X-axis where your method performs well.
- Figure 2. Left plot: your method seems to be outperformed by MLP Aug in the next step. The plot requires an extension to a larger number of samples.
- So far, the presented method performs well only of small datasets and with small neural networks. What is the main limitation of the method?
- What is the time consumption of the proposed method?

---

> ### Author Response · Authors · 2023-11-16
>
> Thank you for the detailed review and positive assessment of our work!
>
>
>
> ____
>
>
>
> We first address the Weaknesses:
>
>
>
> 1. We will update the manuscript and create a separate related work section.
>
>
>
> 2. Please have a look at the Limitations paragraph on p. 9 in our manuscript.
>
>
>
> 3. The major hyperparameters of our model are named in Sections 4.1, 4.2, and 4.3, and are summarized as follows:
>
>
>
> - the depth and width of the model (just like in a vanilla MLP),
>
>
>
> - the bias (True/False),
>
>
>
> - sphere representation (normalized/non-normalized (default), see the last paragraph on p. 2),
>
>
>
> - invariant feature computation (norm/sum).
>
>
>
>
>
>      - Initially, we conducted only a quick hyperparameter search for each task. If the time before the rebuttal deadline allows, we will conduct an ablation study. To this end, what setting specifically is required to be experimented with?
>
>
>
>
>
>
>
> 4. Please see our general comment (the “Low performance” part), the updated Figure 2 in the manuscript, and our responses to the Questions below.
>
>
>
> ____
>
>
>
> Q1. We appreciate the suggestion to reduce CGENN for a fairer comparison, but as an extension of our approach was requested by other reviewers anyway, we compare instead the original CGENN with our extended method modeling higher-order interactions --- the two-point scheme that we mention in our general comment. This add-on does not change the original equivariant feature extraction scheme. It allows our model to get significantly closer to CGENN in the O(5) regression task and outperform it in the two other tasks (see the updated Figure 2 in the manuscript).
>
>
>
> Q2 and Q3. Please refer to our answer to Q1. The experimental setup and datasets for the O(5) Convex Hulls experiment (and O(5) regression) are borrowed from Ruhe et al. (2023), as we stated in Sections 4.2 and 4.3 as well as the second part of the general comment. We only have access to the results of the models, not the full training code, so we have no option to show the competing methods for bigger datasets (except CGENN).
>
>
>
> Q4. The right-most value on the x-axis is the maximum number of training samples in that dataset. As we present in the updated Figure 2, our two-point model outperforms both MLP+Aug and CGENN.
>
>
>
> Q5. As we addressed in the Limitations paragraph (p.9) in the manuscript,
>
> *"More specifically, in the proposed feature propagation, we did not model higher-order interaction between the equivariant features explicitly, which limits the expressiveness of our model"*. We have now addressed it in the general comment (please see the “Low performance” part).
>
>
>
> Q6. Below we provide a comparison of the time complexity of the models in the O(5) Convex Hulls task (the models available to us):
>
>
>
> |DEH          | DEH_PI   |  DEH_$\Delta$_PI  |   CGENN |
> |-------------|-------------|-------------|-------------|
> |    4.3 ms   |   2.7 ms      |   3.1 ms   |      8.2 ms             |
>
>
>
> , where DEHs are our models (see the updated Figure 2).
>
>
>
> (For this, we used the NVIDIA A40.)
>
>
>
> _______
>
>
>
> We sincerely hope this provides clarity and addresses the weaknesses and questions.
>
>
>
> Please let us know should we need to provide any additional details or clarification.

---

### Author Response · Authors · 2023-11-14

We thank the reviewers for their thorough reviews and assessment and helpful feedback!
____
### Low performance

We first address a common criticism concerning comparatively low performance for normal-size and large datasets.

Our neurons describe the relation between point & sphere as given in the paper by Li & Hestenes [1].
Our paper is a study on how to exploit the sphere representation from the paper [1] in a multi-layer network.
The same paper [1] also explains the relation between two points and a sphere, which can be easily put on top of our network, maintaining the original equivariant feature extraction.

The results of the two-point approach are presented in the updated Figure 2 in the manuscript, where DEH_$\Delta$ and DEH_$\Delta$_PI are the accordingly modified versions of DEH and DEH_PI, respectively.

To address another criticism of our experiments having no additional models in the O(3)-invariant action recognition experiment, we added a permutation-invariant variant of CGENN (which we call CGENN_PI), which previously outperformed other models in the other two tasks.
The results are presented in the updated Figure 2 in the manuscript.
 ________

### Training hyperparameters and model sizes

All the models in the comparison have a comparable number of parameters and are trained in the same way (the training hyperparameters are unchanged from https://github.com/DavidRuhe/clifford-group-equivariant-neural-networks), with the same train/val/test split.



|O(3) Action Recognition|       O(5) Regression  | O(5) Convex Hulls |
| Model      | #params | |Model | #params| | Model | #params |
| ------------- | :-----------: | -- | ----------------- | :-----------: | ---- | -------------- | :-----------: |
| DEH_PI    |     7.81K |  |  DEH           |                	391 |  | DEH           |                    55.4K |
|  DEH_$\Delta$_PI |     8.11K |  | DEH_$\Delta$   |            	343 | | DEH_PI    |                     38.2K |
|  CGENN_PI           |    	 8.38K	|  |		 CGENN [2]   |            467 | | CGENN [2]  |          58.8K |
|  MLP (also PI)        |      	 8.31K|  | | | |

The exact details of the architectures of models other than those listed in the table above are unknown to us as we copied their results from Ruhe et al. [2], and they are currently unavailable in the codebase.

___
We will address each reviewer’s comments and questions individually and will update the manuscript accordingly.

____
**References**

[1] Hongbo Li, David Hestenes, and Alyn Rockwood. Generalized homogeneous coordinates for computational geometry. In Geometric Computing with Clifford Algebras, pp. 27–59. Springer, 2001.

[2] David Ruhe, Johannes Brandstetter, and Patrick Forré. Clifford group equivariant neural networks. arXiv preprint arXiv:2305.11141, 2023.

---

### Comment · Area_Chair_Q6W7 · 2023-11-21
**Reviewers: Please respond to authors or update review**

Dear Reviewers,

The discussion phase will end tomorrow.  Could you kindly respond to the authors rebuttal letting them know if they have addressed your concerns  and update your review as appropriate? Thank you.

-AC

---

### Meta-Review · Area_Chair_Q6W7 · 2023-12-11

**Metareview:**

**Summary**
This paper introduces a novel method, Deep Equivariant Hyperspheres, an O(n)-equivariant model generalizing spherical neurons to higher dimensions.  Equivariance is achieved using steerable filter banks constructed using regular n-simplices. The paper lays out the mathematical theory and background supporting the equivariance of the method. Evaluation is performed over an O(3)-invariant classification task and two synthetic O(5)-equivariant tasks.

**Metareview** The method presented is new and a potentially valuable addition to the field of geometric deep learning given that it is distinct from other O(n)-equivariant methods.  The theory supporting the method is worked out in some detail and is correct.  The limited experiments validate the theory and show some advantage for the method in the small data regime, including inference speed. The main limitation of the current paper is that the method is not shown to be effective in practical experiments with real world data (for n>3) against a variety of baselines including other O(n)-equivariant methods.

**Justification For Why Not Higher Score:**

- limited experimental evaluation of method

**Justification For Why Not Lower Score:**

N/A

---

### Decision · Program_Chairs · 2024-01-16

Reject